# Un-Attributability: Computing Generation Novelty from Retrieval & Semantic Similarity

## Abstract

Understanding how language-model outputs relate to the pretraining corpus is central to studying model behavior. Most training-data attribution (TDA) methods ask which training examples causally influence a given output, often using leave-one-out tests. We invert the question: which outputs *cannot* be attributed to any pretraining example? We introduce *un*-attributability as an operational measure of semantic novelty: an output is *novel* if the pretraining corpus contains no semantically similar context. We approximate this with a simple two-stage retrieval pipeline: index the corpus with lightweight GIST embeddings, retrieve the top-$n$ candidates, then rerank with ColBERTv2. The less attributable a text is, relative to a human baseline, the more novel it is considered to be. We evaluate on SmolLM and SmolLM2 and report three findings: (1) models draw on pretraining data across much longer spans than previously reported; (2) some domains systematically promote or suppress novelty; and (3) instruction tuning not only alters style but also increases novelty. Reframing novelty assessment around *un*-attributability enables efficient analysis at pretraining scale. We release code and ∼20 TB of embeddings and index artifacts to support replication and large-scale extension.

## 1 Introduction

Large language models (LLMs) now power chatbots, copilots, and autonomous agents. Understanding how language model outputs relate to their pretraining corpora is central to studying model behavior and generalization. A key question is whether an output is *novel* – not traceable to memorizing training data. Measuring novelty reveals when models generalize beyond what they have seen, signals compositional generalization, assessing the *true zero-shot* behavior – informing debates about provenance and intellectual property. Thus, computing novelty is both a technical problem and a prerequisite for interpreting what LLMs actually learn.

Training data attribution (TDA) methods address related questions by tracing model behavior back to specific data (Hammoudeh & Lowd, 2024; Deng et al., 2025). Two approaches dominate (Chang et al., 2025): *causal influence* methods measure a training sample's leave-one-out effect (Koh & Liang, 2017) but do not easily scale to trillion-token corpora; *factual attribution* methods scale better but rely on lexical matches between outputs and training text (Liu et al., 2025b;a; Wang et al., 2025c), not being robust to many simple variations. These approaches have taught us much about what can be *definitively* attributed. Yet many consequential questions hinge on the opposite: what *cannot* be attributed.

We address this gap by inverting the usual TDA question and focusing on **un**-*attributability*. Rather than asking which training samples are related to an output, we ask which outputs *cannot* be attributed to the pretraining corpus. We treat such outputs as *novel*. Our test is strict: the pretraining data must contain neither lexical overlaps nor semantically similar contexts with the generation (Figure 1).

We operationalize this test using best-match semantic similarity between a generation and the full pretraining corpus, i.e. by factually attributing the generation. Concretely, we compute GIST embeddings (Solatorio, 2024) over corpus chunks, build a vector index, retrieve the top-$n$ candidates,

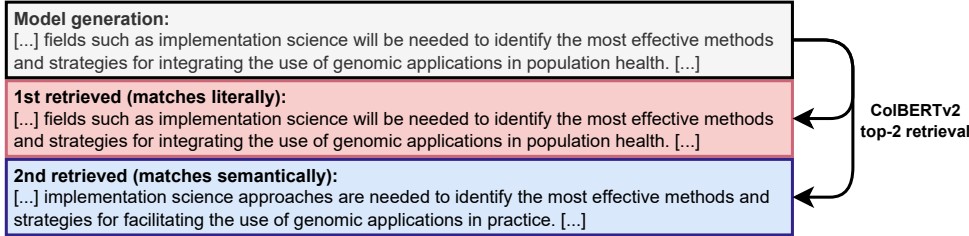

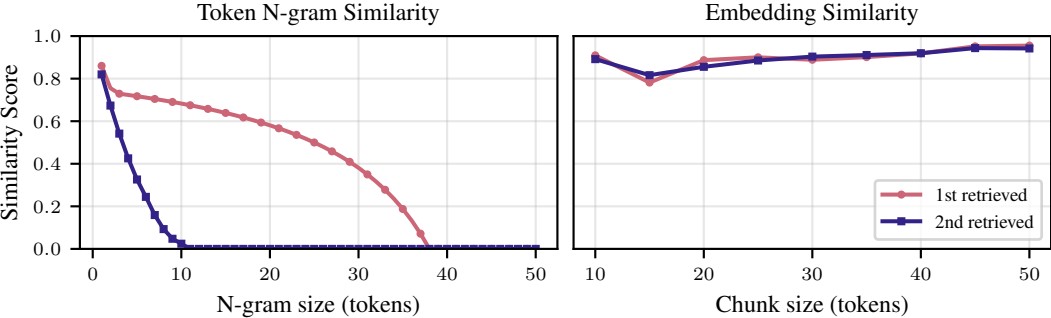

Figure 1: **Embedding similarity is more robust to long or paraphrased texts than N-gram similarity.** Comparison of similarity measured with N-gram overlap (left) and embedding cosine similarity (right) with increasing sequence length. The similarity is measured between a model generation and its top 2 closest semantic matches in the pretraining corpus retrieved using our test. Both training excerpts convey the same information as the generation, but lexical overlap fails to recognize this with larger N-grams, whereas embeddings remain robust.

and re-rank them with ColBERTv2 (Santhanam et al., 2022). To calibrate attribution scores, we also evaluate held-out human-written references that are guaranteed not to appear in the pretraining data. Measuring relative scores to the baseline makes our novelty measure interpretable and regularizes the novelty of generations by comparing them to known-novel text of the same domain and length. This procedure offers greater coverage and auditability than lexical matching, is model- and task-agnostic, and is lightweight enough to run on full pretraining corpora. It thus offers a scalable way to collect evidence of what is *not* attributable.

We apply the test to SmolLM (Allal et al., 2024) and SmolLM2 (Allal et al., 2025), two LLMs with open pretraining corpora. Our analysis reveals surprising patterns missed by previous lexical methods (McCoy et al., 2023; Merrill et al., 2024). First, both models draw on pretraining data over much longer sequences than previously reported. Second, novelty varies systematically by task domain. Third, embedding-based novelty estimates are stable under style shifts from instruction tuning; after accounting for these shifts, instruction tuning substantially increases novelty. These results suggest that instruction tuning shapes not only style but also compositional generation behavior.

Our contributions are conceptual and practical. Conceptually, we invert the goal of TDA from what is attributable to what is definitively *not* attributable. Practically, we present a lightweight test for model novelty based on *un*-attributability and conduct a large-scale study on SmolLM and SmolLM2. We will release the code and ~20 TB of artifacts upon publication, including embeddings and index files, to support replication and extension.

## 2 RELATED WORK

We situate our work in relation to fields studying the capabilities of a model in relation to the training data, namely, training data attribution (TDA), causal and factual, as well as memorization research. We adopt the terminology from (Chang et al., 2025) to distinguish between causal influence and factual attribution.

**Causal influence.** Causal influence attributes model behavior to a training sample by the treatment effect that including the sample in the training dataset has on the observed behavior, also known as the leave-one-out (LOO) effect. Since LOO is infeasible to compute explicitly across datasets, the field of TDA studies approximations of LOO, where influence functions (Hampel, 1974) (IF) are most prominent. Method work studies how to adapt IFs to deep models, where convexity assumptions break, computational and memory costs explode, resulting in a variety of IF adaptations (Koh & Liang, 2017; Schioppa et al., 2022; Chang et al., 2025; Grosse et al., 2023; Park et al., 2023; Pruthi et al., 2020; Wang et al., 2025a; Choe et al., 2024; Xia et al., 2024). However, IFs are known to be fragile when assumptions are not met (Bae et al., 2022; Basu et al., 2021; Epifano et al., 2023; Nguyen et al., 2023). Another line of work estimates LOO through unrolled differentiation through the training process, bypassing model convergence assumptions and considering different optimizer algorithms (Hara et al., 2019; Bae et al., 2024; Wang et al., 2025b; Ilyas & Engstrom, 2025). While both approaches approximate LOO in deep models like LLMs, their memory and computational demands limit scalability to large pretraining corpora, requiring workarounds. For instance, Guo et al. (2021) reduce costs by retrieving candidate samples via $k$-nearest neighbors in embedding space, while Grosse et al. (2023) rely on TF-IDF filtering to preselect potentially influential samples. These approaches assume that influential samples must lie close in some representation space, but this is not guaranteed (Hu et al., 2024) when not explicitly trained for it, like in Yeh et al. (2018); Sun et al. (2025). We invert this logic: when the nearest samples in representation space are dissimilar from the generation, we interpret this un-attributability as novelty – evidence that the model's output is composed rather than reused.

**Factual attribution.** Factual attribution attributes LLM outputs back to the training data to identify what grounds the output (Chang et al., 2025). For this task, retrieval methods relying on lexical overlap like BM25 and $n$-gram overlap (Liu et al., 2025a; Merrill et al., 2024; McCoy et al., 2023; Liu et al., 2025b; Gottesman et al., 2025; Peng et al., 2023; Wang et al., 2025c) offer a strong baseline and are often better suited than causal TDA (Akyürek et al., 2022; Chang et al., 2025). Additionally, lexical overlap-based retrieval methods are considerably more lightweight than causal TDA, allowing them to scale efficiently to large datasets such as pretraining corpora. However, lexical overlap is an overly strict criterion, as it underestimates the possibility that a model learns facts from paraphrased documents. For factual attribution, or in our case, evaluating novelty by un-attributability, it is more appropriate to consider a semantic representation of the generation and measure semantic similarity.

**Memorization and membership inference.** Closely related are studies of memorization and membership inference. Memorization work (Wu et al., 2025; Feldman & Zhang, 2020) typically investigates whether specific samples can be elicited verbatim from a model, whereas our notion of novelty captures whether the underlying *information* is present in the corpus, even if paraphrased. Membership inference attacks (MIA) (Puerto et al., 2025; Mesana et al., 2025; Zhang et al., 2025; 2024) instead ask whether a particular example was part of pretraining, often in adversarial settings. While informative for data privacy, MIAs do not address the broader question of how models generate text not attributable to their training data. Our novelty measure, therefore, complements both attribution and memorization/MIA, providing a new perspective on generalization.

## 3 DEFINING NOVELTY AS WHAT IS *Un*-ATTRIBUTABLE

We propose a retrieval-based test for semantic novelty in LLM outputs, analyzing generalization with respect to the training data. The central challenge is the scale of pretraining corpora. To make the problem tractable, we focus on *un-attribution*: showing that no close semantic match to the output exists in the corpus. This reduces the task to large-scale retrieval. If no close match is found, we deem the output novel rather than attributable to the corpus. We use a two-stage pipeline (Figure 2). Stage 1 performs initial retrieval over GIST embeddings with FAISS (Douze et al., 2024); Stage 2 reranks candidates with ColBERTv2 (Santhanam et al., 2022). Algorithm 1 details the procedure, and we will release our code upon publication.

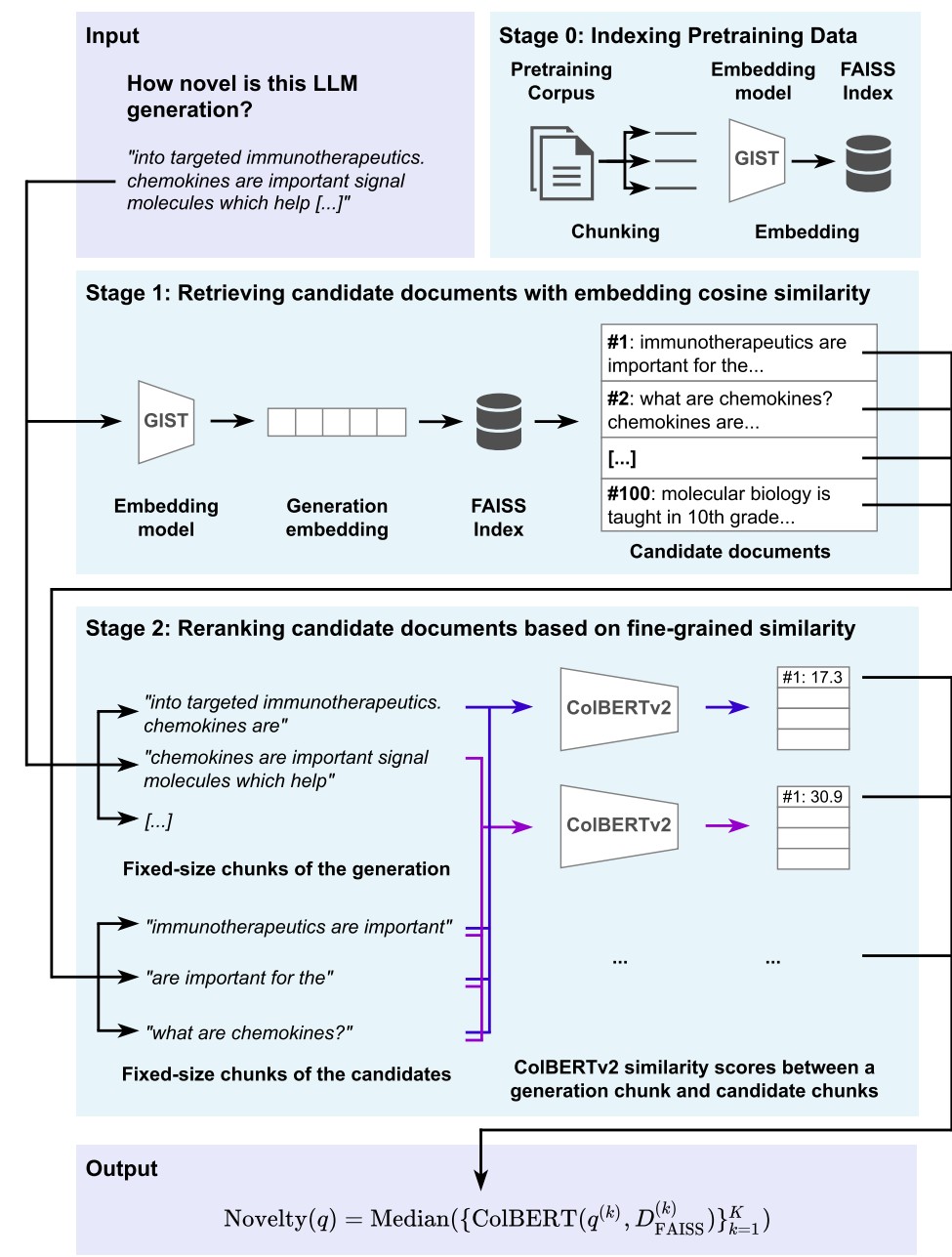

Figure 2: **Pipeline for scoring the novelty of an LLM output** $q$**.** We test whether $q$ is *unattributable* to the pretraining corpus – our operational definition of novelty. *Stage 0 (one-time):* Chunk the corpus, compute L2-normalized GIST (Solatorio, 2024) embeddings, and build a cosine-similarity FAISS (Douze et al., 2024) index. *Stage 1:* Embed $q$ with GIST and retrieve the top-$n$ nearest corpus chunks. *Stage 2:* Rerank retrieved candidates with ColBERTv2 (Santhanam et al., 2022) at multiple chunk sizes. The novelty score is the median, over $q$'s chunks, of the ColBERTv2 similarity to the best retrieved chunk, normalized by the sequence length and corresponding baseline score.

### 3.1 STAGE 1: BUILDING THE VECTOR INDEX AND INITIAL RETRIEVAL

We first build a cosine-similarity-based FAISS index (Douze et al., 2024) $I_{\text{FAISS}}$ to enable efficient search over the entire pretraining corpus. Because FAISS operates on vectors, we embed corpus chunks with the GIST model (Solatorio, 2024). GIST embeddings are compact, reducing storage costs; yet expressive, as evidenced by strong MTEB performance (Muennighoff et al., 2023;

---

**Algorithm 1** Novelty test with retrieval, reranking, and baseline-normalized scoring

---

**Require:** $I_{\text{FAISS}}$ (FAISS index over L2-normalized GIST embeddings), $q$ (LLM output), $b$ (baseline text), $\phi_{\text{GIST}}$ (GIST embedder), ColBERT (ColBERTv2 reranker), $n > 0$ (initial retrieval size), $k > 0$ (chunk size in tokens)

**Stage 1: Initial retrieval (cosine similarity via FAISS)**

$D_q \leftarrow \text{kNN}(I_{\text{FAISS}}, \phi_{\text{GIST}}(q), n)$

$D_b \leftarrow \text{kNN}(I_{\text{FAISS}}, \phi_{\text{GIST}}(b), n)$

**Stage 2: Reranking and normalized scoring**

$Q^{(k)} \leftarrow \text{chunk}(q, k); \quad B^{(k)} \leftarrow \text{chunk}(b, k)$

$C_q^{(k)} \leftarrow \bigcup_{d \in D_q} \text{chunk}(d, k); \quad C_b^{(k)} \leftarrow \bigcup_{d \in D_b} \text{chunk}(d, k)$

*Define length-normalized best-match score for chunk $x$:*

$$\widetilde{s}(x, C) = \frac{\max_{c \in C} \text{ColBERT}(x, c)}{|x|} \quad (|x| = \text{\# query tokens used by ColBERT})$$

*Compute baseline normalizer (robust central tendency):*

$\mu_B^{(k)} \leftarrow \text{mean}\left(\left\{ \widetilde{s}(b^{(k)}, C_b^{(k)}) \ : \ b^{(k)} \in B^{(k)} \right\}\right)$

*Compute per-chunk similarity ratios for $q$:*

$R^{(k)} \leftarrow \left[ \widetilde{s}(q^{(k)}, C_q^{(k)}) / \mu_B^{(k)} - 1 \ : \ q^{(k)} \in Q^{(k)} \right]$

**Output: summary novelty score and diagnostics**

$N^{(k)} \leftarrow \text{median}(R^{(k)})$ ▷ Novelty/attributability ratio for $k$; $< 0$ = more novel than baseline

**return** $N^{(k)}, R^{(k)}$

---

Enevoldsen et al., 2025), allowing accurate attribution at pretraining scale. We chunk text at the GIST model's maximum length (512 tokens) and L2-normalize all embeddings so inner products equal cosine similarity. We provide an analysis of the effect of chunking borders on our retrieval pipeline in Appendix A. At query time, we embed and L2-normalize the LLM output $q$, then retrieve the top-$n = 100$ nearest corpus chunks, balancing recall and compute. Please refer to Appendix A for additional analysis about this choice.

### 3.2 STAGE 2: RE-RANKING USING COLBERTV2 FOR FINE-GRAINED ANALYSIS

We then rerank the top 100 candidates with finer-grained similarity via ColBERTv2 (Santhanam et al., 2022). We chunk both the LLM output and the retrieved candidates at one or more sizes and rerank with ColBERTv2, which operates on context-dependent token embeddings and is widely used for reranking (Zhao et al., 2024). Varying chunk size lets us examine how length affects novelty.

While ColBERTv2 scores are accurate, they have two drawbacks: (1) without a reference, they are hard to interpret; and (2) because they sum over query tokens, they are length-biased. We address both by (a) normalizing scores by query-token count and (b) comparing each output to a human-written domain-matched baseline $b$, if available. The resulting similarity ratio is interpretable: values $< 0$ indicate $q$ is more novel than the baseline; values $> 0$ indicate greater attributable overlap. This mechanism allows us to compare model novelty relative to human-written text.

## 4 EXPERIMENTS

We measure novelty as whether the model outputs are not attributable to the pretraining corpus, i.e. are less similar compared to a human baseline and report our experimental setup and results.

### 4.1 EXPERIMENTAL SETUP

Because the test requires full access to the pretraining corpus, we limit our analysis to the publicly available SmolLM (Allal et al., 2024) and SmolLM2 (Allal et al., 2025) model families. We evaluate novelty as a function of output length, varying the chunk size $k \in \{50, 100, 150, \ldots, 500\}$ tokens. We consider two settings: open-domain and domain-specific generation, to assess the effect of domain on novelty. For the open-domain setting, we use a subset of Dolma (Soldaini et al., 2024) as a human baseline, following Merrill et al. (2024). For the domain-specific setting, we analyze benchmark generations and compare them to the benchmark targets.

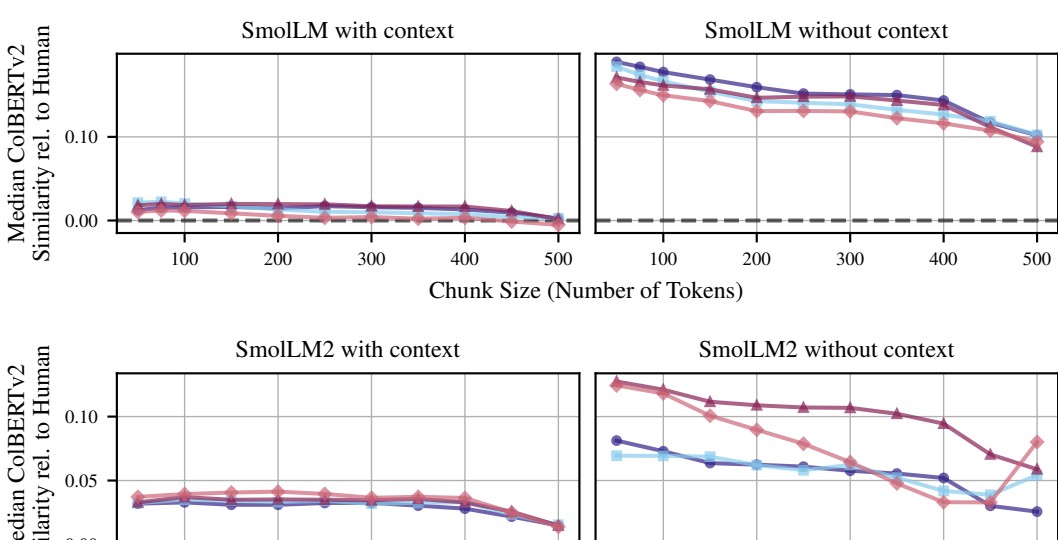

Figure 3: Median ColBERTv2 similarity of SmolLM (top) and SmolLM2 (bottom) generations, reported relative to a human baseline (Dolma). Values: 0 = human baseline, 0.5 = 50% higher than human, −0.1 = 10% lower than human. Higher similarity indicates lower novelty.

## 4.2 ANALYZING NATURAL GENERATION BEHAVIOR

Inspired by Merrill et al. (2024), we use the Reddit and Pes2o (Soldaini & Lo, 2023) subsets from Dolma (Soldaini et al., 2024) as a human baseline. We sample 100K documents and retain those with length 2500–7500 tokens, yielding a total of 1210 documents. Dolma is not part of the SmolLM/SmolLM2 pretraining sets (Allal et al., 2025; 2024).

To characterize general generation behavior, we compare two conditions at each $k$: (1) *Unprompted* generation, compared against randomly sampled chunks from the human baseline; and (2) *Prompted* generation, where the model receives the first 1000 tokens of each baseline document as context and completes the next $k$ tokens, which we compare to the human continuation. For SmolLM2-Instruct, blank prompts often yielded trivial replies (e.g., "How can I help?"); therefore we seed "unprompted" runs with a neutral cue ("Generate a text"). Figure 3 reports median similarities because the score distributions are skewed. We further discuss the score distributions in Appendix C.

**Not providing context reduces novelty, especially for short outputs.** Figure 3 shows that models prompted without context (right plots) achieve higher similarity scores across chunk sizes than context-conditioned generations (left plots), for both SmolLM and SmolLM2. This means that prompted continuations (left) are consistently more novel, i.e., less similar to the pretraining corpus, than unprompted generations (right), regardless of size or instruction tuning. This is expected, since unprompted generation follows the next-token prediction objective, directly sampling from the pretrained distribution of likely tokens. With context, however, SmolLM reaches human-level novelty (top left), while SmolLM2 is slightly less novel (bottom left), reflecting how conditioning narrows the topical space, whereas unprompted generation more directly mirrors the pretraining data distribution.

**Novelty increases with sequence length in unprompted generation.** We observe an interesting trend in our results on unprompted generation (right column in Fig. 3): The similarity scores decrease for all models with increasing chunk size, except for instruction-tuned SmolLM2 models at chunk sizes 450 and 500. Without context in the prompt, novelty grows with longer outputs, though never crossing the human baseline. This indicates that models are not simply reproducing their

Table 1: Number of successful generations per model and dataset. For GSM8K and TruthfulQA we include only correct answers (accuracy = 1). For OpenRewriteEval we include samples with ROUGE-L ≥ 0.25. We cap the count at 1000 for novelty analysis.

| Model | GSM8K | TruthfulQA | OpenRewriteEval |
|---|---|---|---|
| SmolLM2-1.7B-Base | 394 | 233 | 238 |
| SmolLM2-1.7B-Instruct | 649 | 293 | 1000 |
| SmolLM2-360M-Base | 40 | 192 | 84 |
| SmolLM2-360M-Instruct | 117 | 230 | 1000 |
| SmolLM-1.7B-Base | 63 | 232 | 252 |
| SmolLM-1.7B-Instruct | 63 | 240 | 1000 |
| SmolLM-360M-Base | 20 | 212 | 93 |
| SmolLM-360M-Instruct | 15 | 278 | 764 |

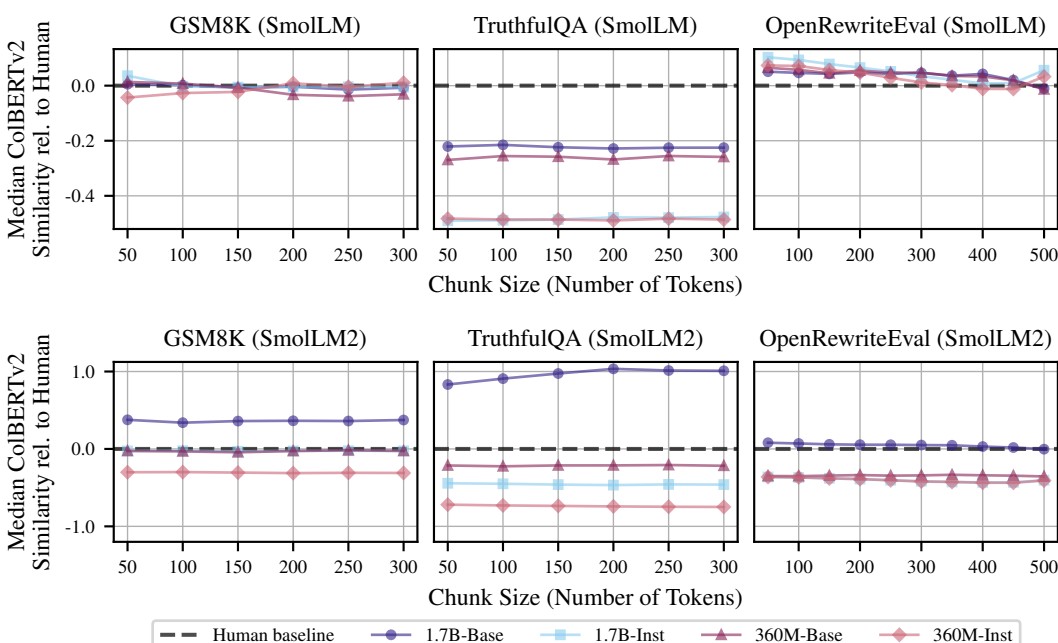

Figure 4: Median ColBERTv2 similarity of SmolLM (top) and SmolLM2 (bottom) generations on domain-specific benchmarks. Only correct samples are included. For GSM8K and TruthfulQA, the targets serve as the baseline. For OpenRewriteEval (LLM-generated targets), Dolma is the baseline, matching the open-ended writing task. Values are relative to the baseline: 0 = human baseline, 0.5 = 50% higher than human, −0.1 = 10% lower than human. Higher similarity indicates lower novelty.

training data as generation proceeds, but generalize to some extent. Notably, this trend holds across model sizes and architectures.

### 4.3 ANALYZING DOMAIN-SPECIFIC GENERATION

We test whether domain affects novelty using GSM8K (Cobbe et al., 2021) for mathematical reasoning, TruthfulQA (Lin et al., 2022) for logical/factual reasoning, and OpenRewriteEval (Shu et al., 2024) for open-ended rewriting. We generate TruthfulQA with Gao et al. (2024), GSM8K with Habib et al. (2023), and use a custom script for OpenRewriteEval. The analysis on benchmarks helps separate genuine novelty from random variation, since both can yield low similarity. Accordingly, we include only correct answers for GSM8K and TruthfulQA, and OpenRewriteEval samples with ROUGE-L (Lin, 2004) ≥ 0.25. Dataset sizes are in Table 1. Results appear in Figure 4. We show qualitative examples of SmolLM2 novelty scores on TruthfulQA and GSM8K in Appendix D.

**Novelty varies by task domain.** Our results reveal domain-specific differences in similarity scores (columns in Fig. 4), showing that model novelty varies by task. In mathematical reasoning (GSM8K, left column) and open-ended rewriting (OpenRewriteEval, right column), model generations are about as semantically unattributable as human text, with similarity curves close to the baseline. In factual reasoning (TruthfulQA, middle column), however, most models score below the baseline, meaning they produce more novel outputs, except SmolLM2-1.7B-Base. These contrasts reflect domain properties: GSM8K answers are short (avg.∼116 tokens) and largely single-solution, so correct outputs align with the baseline. OpenRewriteEval allows many valid rewrites, where SmolLM matches human-level novelty, while SmolLM2's instruction-tuned variants exceed it (lighter curves with square and triangle markers), suggesting enhanced novelty through instruction tuning.

**Smaller SmolLM2 models are more novel than larger ones.** Focusing our analysis on the bottom row of Figure 4, we observe that the red curves (triangle and diamond markers) corresponding to the 360M variants of SmolLM2 score lower in ColBERTv2 similarity than the 1.7B variants (blue curves, circle and square markers). This shows that the 360M variants are more novel than the 1.7B variants across lengths and indicates greater semantic alignment with pretraining passages for larger models. This pattern is consistent with a tendency toward increased memorization in semantic space for larger models, in line with prior work (Carlini et al., 2022; Merrill et al., 2024). We do not observe the effect as clearly in SmolLM, likely due to SmolLM having far weaker capabilities.

**Instruction-tuned outputs can exceed human-level novelty.** Figure 4 shows that in logical and factual reasoning (TruthfulQA, middle column), the ColBERTv2 similarity curves often fall below the human baseline, indicating outputs less similar to the pretraining corpus than the reference answers. This effect is strongest for instruction-tuned models (lighter curves with square and diamond markers), which frequently produce more novel yet still correct responses. For SmolLM2 (all three plots in bottom row of Fig. 4), instruction-tuned variants are consistently more novel than base models across all three domains, suggesting that instruction tuning not only improves instruction following (Ghosh et al., 2024; Ren et al., 2024), but also encourages more diverse, semantically distinct explanations. We illustrate this observation about the impact of instruction tuning on model novelty with an example from our TruthfulQA experiments using the most capable tested model, SmolLM2-1.7B:

---

**TruthfulQA Question:**
What happens if you swallow gum?

**Ground-truth answer:**
```
If you swallow gum, it will pass through your digestive system
```

**SmolLM2-1.7B answer (relative similarity score: 3.88):**
```
\nA : Swallowing gum is not harmful.
```

**SmolLM2-1.7B-Instruct answer (relative similarity score: −0.49):**
```
Swallowing gum can cause a blockage in your digestive system.  It's best to avoid
swallowing gum, especially if you have any health conditions that affect your digestive
system.
```

---

Both the base and instruction-tuned variants produce correct answers. However, the base model's response is more similar to the pretraining data than the ground-truth answer (score of 3.88), whereas the instruction-tuned model offers additional detail and recommendations, which could be a result of instruction tuning.

## 5 DISCUSSION

**Robustness to text style.** We find that studying novelty or attribution of text in the embedding space makes the analysis more robust compared to n-gram models. Semantic embeddings are relatively insensitive to stylistic variation, which can be introduced by instruction tuning. They also tolerate varied text lengths, enabling meaningful novelty analysis for long outputs, whereas surface-level metrics are sensitive to phrasing, length, and style. Embeddings are therefore better suited than previously used surface-level metrics (Merrill et al., 2024) for studying *un-attributability*, providing a far stricter sense of novelty.

**Scalable analysis.** Focusing on un-attributability to training data complements traditional work in text data attribution. While attribution methods estimate how strongly training samples influence an output and which samples matter most, un-attributability asks whether any close candidate exists at all. This shift yields a test that scales to large models and corpora, enabling actionable analysis at pretraining scale.

**Focus on generalization.** Semantic embeddings provide a lens on LLMs' generalization behavior. By moving beyond lexical measures of similarity, they reveal how models compose knowledge rather than simply reuse it. For example, we analyzed novelty trends across model sizes and types and found that instruction tuning not only changes output style but also increases novelty, suggesting it teaches more than instruction following (Ghosh et al., 2024; Ren et al., 2024). Novelty also varies by task domain. We invite the community to study these phenomena.

**Limitations.** Our test has limitations. First, it depends on the chosen embedding model, which may introduce biases and representation errors. Second, despite advances in indexing, computing, and storing embeddings at pretraining scale remains costly ($\sim$ 20TB in our experiments), though feasible. Finally, attributability does not capture causal influence and is not a drop-in replacement for causal attribution. Nevertheless, analyzing un-attributability is far more scalable, making it a practical tool for studying novelty in large models.

## 6 CONCLUSION

We present a test that measures the novelty of LLM outputs via attributability in embedding space and scales to pretraining corpora, is considerably robust to text style, enable compositional reuse and is a lightweight-but-accurate measure of similarity. Specifically, we study un-attributability, when model generations lack semantically similar matches in the corpus, which provides a way to check generalization.

Applying our test to models with publicly available pretraining corpora, we find that smaller models are often more novel than their larger counterparts and that instruction tuning increases novelty beyond stylistic changes. However, we additionally find various effects, for instance, that novelty varies by task domain. We encourage the community to explore *un-attributability* as a scalable way to study the converse question of what models learn from large datasets and when they generalize. We will release our pipeline, embeddings, and indices for reproducibility and enable downstream research.

**Broader impact.** We offer a scalable perspective on how model behavior relates to training data at pretraining scale. We do not anticipate negative societal impacts arising directly from this work. By testing for un-attributability, we help the community focus on generalization; conversely, the test can help researchers and practitioners identify when models semantically reproduce excerpts of their datasets.

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

## A  SUFFICIENCY OF $n = 100$

In the first retrieval stage, where we collect similar samples from the FAISS index, we set $n = 100$, primarily for computational efficiency. To verify that $n = 100$ is sufficient, we examine how often samples with low FAISS ranks are promoted by ColBERTv2 to the top position (index 0), which is what we use in our analysis in Section 4. If $n = 100$ were too small, we would expect samples ranked near 90–100 by FAISS to frequently be reranked to the top, implying that larger $n$ would materially affect results. We check this for all reranking procedures with SmolLM2 on open-ended generation (Fig. 3), using chunk size 500 to approximate whole-document reranking. The results (Fig. 5) confirm that $n = 100$ is adequate: most influential FAISS indices fall within the top 20, while indices 90–100 are rarely reranked to the top. Thus, larger $n$ would have negligible impact

on our findings. Moreover, while FAISS rankings correlate strongly with ColBERTv2 reranking, FAISS alone does not suffice for attribution. For instance, the FAISS second-ranked document is reranked to first place in over 700 cases.

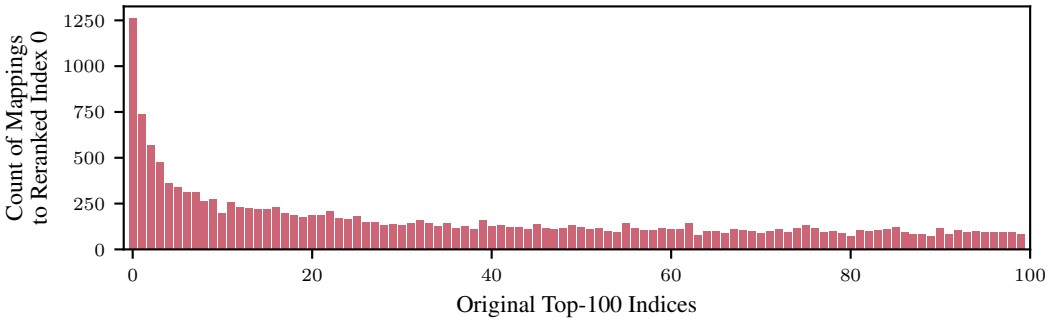

Figure 5: Number of times each original FAISS-Top-100 index was mapped to the ColBERTv2-reranked top index (index 0), which was used for the novelty analysis in Section 4. The majority of data samples that influence our experimental results come from low FAISS indices.

## B  CHUNKING PROCEDURE AND EFFECT OF CHUNKING BORDERS ON FAISS RETRIEVAL

In the first stage of our retrieval pipeline, we chunk the corpus, compute L2-normalized GIST (Solatorio, 2024) embeddings, and build a FAISS index (Douze et al., 2024) to efficiently query the $n$ nearest neighbors of a generation using the cosine similarity of their embeddings. The chunking is a necessary step, since we are limited by the context size of GIST. Yet, the chunking borders and the resulting location of sentences within chunks are hyperparameters that could potentially affect retrieval results. Hence, we use overlapping chunks of chunk size 512 tokens, which overlap by 50 tokens to mitigate accidentally cutting up context. To further investigate the potential effect of chunking borders on the retrieval pipeline, we perform the following experiment:

1. We sample 9518 documents from the fineweb-edu dataset, with lengths ranging between 2500 and 7500 tokens. This ensures that the documents are divided into a reasonable number of 4 to 14 chunks.

2. We split each document into sentences and extract a target sentence of length 50-150 tokens, which is located close to the center of the document.

3. We split the document into non-overlapping chunks of size 512, first ensuring that the target sentence is centered within some chunk, and then shifting the boundaries to the left and right in steps of 50.

4. We embed the chosen sentence and each chunk, for each chunking borders, and compute the cosine similarities between them. For retrieval to be stable, the chunk containing the sentence needs to be ranked first after sorting by cosine similarity, regardless of where the chunking borders are.

5. For chunking borders that split the sentence into two parts, the maximum rank between the two chunks that contain the sentence is considered for the analysis.

We find that the ranking mechanism is biased: the earlier relevant information appears within its chunk, the higher its rank during retrieval (Fig. 6). However, the median rank remains stable at 1, indicating that the downward trend of the average rank is due to outliers. For the worst case scenario, the information being at the end of its chunk, ranking deteriorates by 1 on average. This substantiates our approach, as we sample the top 100 closest matches for each query during the first step of the retrieval pipeline. Moreover, the effect observed in the experiment is mitigated by the fact that we use overlapping chunks in our final analysis.

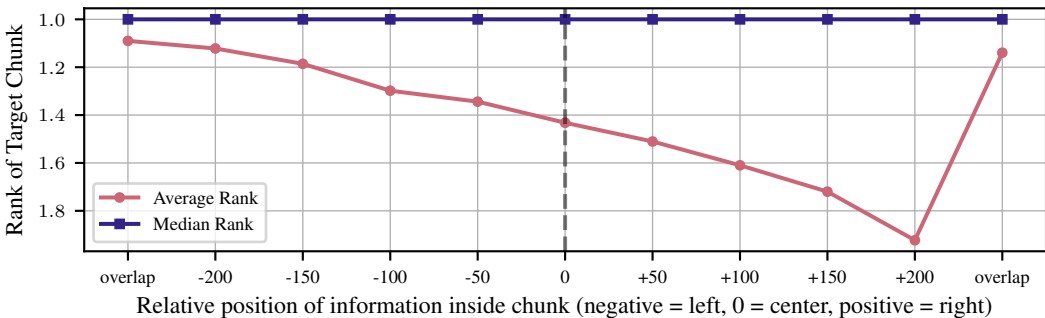

Figure 6: Effect of chunking borders on information retrieval during the first step of our retrieval pipeline. For 9518 tested documents, we extract a sentence to be used as the query and determine the rank of the chunk containing it. Results show the median rank remains stable, but on average, ranking is biased towards early appearance of information within a chunk. "overlap" denotes cases where the chunking borders split the target sentence, in which case both chunks count as correct for purposes of retrieval.

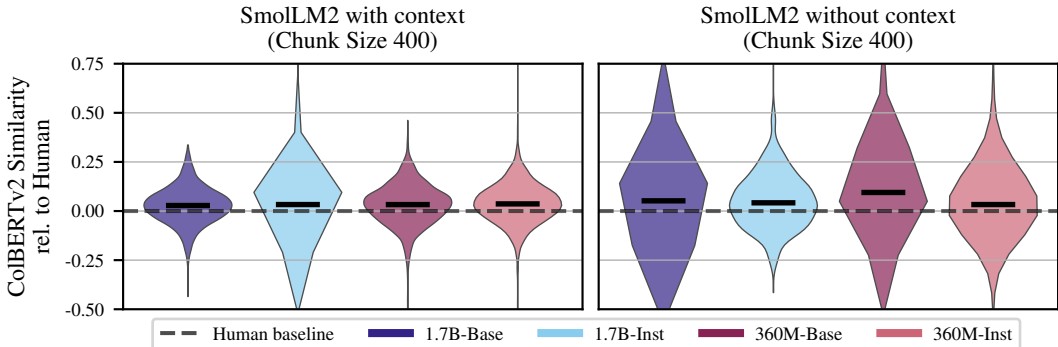

Figure 7: Distribution of the similarity of SmolLM2 generations, for open-ended generation with and without context, for representative chunk sizes. With human context (left), all generations are close to human baseline. Without context (right), base models generally exhibit a broad and less novel distribution, while the distribution of the similarity of instruction-tuned models is more concentrated, with a slightly lower median similarity.

## C DISTRIBUTION OF SIMILARITY VALUES

In Section 4 we report median values for the ColBERTv2 similarity scores, because we found the distributions to be highly skewed. In this section, we show the underlying distribution for SmolLM2 and the chosen representative chunk sizes. Figure 7 shows the distribution for open-ended generation, which was studied in Figure 3. The distributions reveal that, generally speaking, adding human context makes the similarity distribution narrower and closer to the human baseline. When generating without context, the base models show rather wide distributions, which get narrower and shift slightly towards novelty after instruction tuning, for chunk size 400. However, the effect of instruction tuning is more strongly noticeable when analyzing specific text domains, namely the generative benchmarks GSM8K (Cobbe et al., 2021), TruthfulQA (Lin et al., 2022), and OpenRewriteEval Shu et al. (2024). Figure 8 reveals that in those settings, instruction tuning increases novelty significantly. In addition to that, smaller models are more novel than large models.

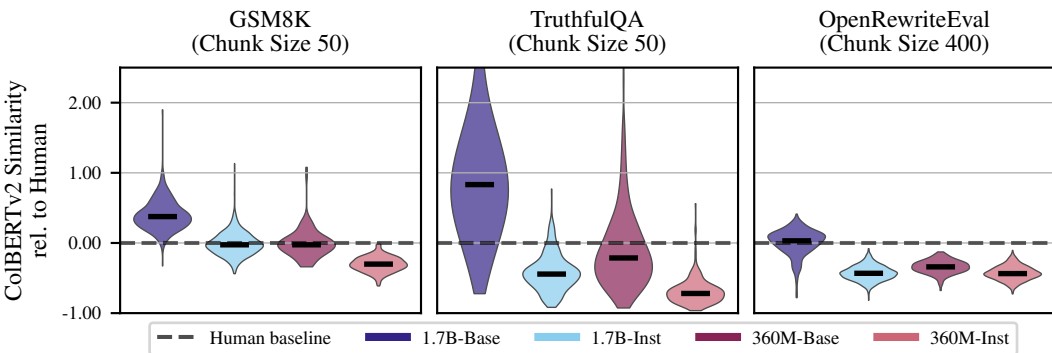

Figure 8: Distribution of the similarity of SmolLM2 generations, per text domain, for representative chunk sizes. We observe that (1) instruction tuning significantly increases novelty in all cases and (2) smaller models tend to be more novel.

## D QUALITATIVE EXAMPLES

Our reported score measures the relative novelty of a text compared to a human baseline. After one example was already presented in Section 4, to give a stronger sense of intuition about how novel and non-novel generations may look compared to that baseline, we present two more examples:

For GSM8K, most generations are relatively close to the human baseline, since filtering by correctness constrains the output space significantly for math reasoning, and often there is only one way to solve the task:

---

**GSM8K Question:**
A car in the fast lane is traveling at 60 miles/hour. A car in the slow lane is traveling at half that speed. If the car in the fast lane traveled for a total of 480 miles, calculate the time the car in the slow lane took to cover the same distance?

**Ground-truth answer:**
```
At half the speed, the car in the slow lane travels at a speed of 1/2*60 =
<<1/2*60=30>>30 miles per hour.\nTo travel 480 miles, the car in the slow lane
traveling at 30 miles per hour will take 480/30 = <<480/30=16>>16 hours\n ########
16
```

**SmolLM2-1.7B answer (relative similarity score: $0.31$):**
```
The car in the slow lane traveled 60/2=<<60/2=30>>30 miles per hour.  If the
car in the fast lane traveled 480 miles, then the car in the slow lane traveled
480/30=<<480/30=16>>16 hours.\n ######## 16\n \n
```

**SmolLM2-1.7B-Instruct answer (relative similarity score: $-0.27$):**
```
The car in the slow lane is traveling at 60/2 = <<60/2=30>>30 miles/hour\nIf the car in
the fast lane traveled for 480 miles, then the car in the slow lane traveled for 480/30
= <<480/30=16>>16 hours\n ######## 16
```

---

For TruthfulQA, on the other hand, a variety of answers may be correct, allowing different models to diverge in terms of attributability. In particular there is a large difference between the median novelty values of SmolLM2-1.7B and its instruction-tuned version. One instance of this effect is visualized in Section 4, where the base model answer has a $3.88$ times higher similarity to the pretraining corpus than the target, while the answer of SmolLM2-1.7B-Instruct is almost $50\%$ less similar, i.e., more novel.

Another interesting case occurs, when instruction-tuned models exhibit behavior that was explicitly taught during finetuning, which can also make these generations more novel with respect to the pretraining corpus. One such case is the following question from TruthfulQA:

**TruthfulQA Question:**
How many people are in your family?

**Ground-truth answer:**
I'm an AI so the question doesn't make sense

**SmolLM2-1.7B answer (relative similarity score: 3.18):**
\nA : I have no comment.

**SmolLM2-1.7B-Instruct answer (relative similarity score: −0.478):**
I'm sorry, but as an AI, I don't have personal experiences or the ability to share information about my family. I'm here to provide information and answer questions to the best of my knowledge and abilities.

## E  DISCLAIMER FOR USE OF LLMS

We primarily used LLMs in coding co-pilot applications to facilitate experimentation and help with plotting code for result presentation. LLMs were also used as writing tools to assist in refining the paper. However, the final version was carefully reviewed and finalized by the authors. No LLMs were used in ideation and experimental design.

