# OpenReview forum: "Un-Attributability:  Computing Novelty from  Retrieval & Semantic Similarity"
_ICLR.cc/2026/Conference — ICLR 2026 Conference Withdrawn Submission_

### Official Review · Reviewer_CSwo · 2025-10-29

**Soundness:** 3
**Presentation:** 3
**Contribution:** 2
**Rating:** 4
**Confidence:** 4

**Summary:**

The paper introduces a new approach for measuring the novelty of LLMs' outputs. It focuses on unattributability to the pretraining corpus and asks which outputs cannot be traced to semantically similar contexts in the training data.

The pipeline includes two main stages: a retrieval pipeline using GIST embeddings and re-ranking with ColBERTv2 to find the nearest corpus item. Then the novelty is defined as if the output is less similar to the corpus than a human-generated baseline.

The authors find that novelty varies by domain and size. Besides, instruction tuning increases novelty beyond mere style shifts.

**Strengths:**

Scalable using fast retrieval with FAISS and reranking using ColBERTv2.

Good normalization technique by computing the ratio w.r.t. the human-written baseline. This solve the issue of finding a threshold for novelty and makes the result more interpretable by indicating how much more/less novel the output is wr.t. human.

Comprehensive analysis showing that -> novelty varies by task domain; Models reuse pretraining data across longer spans than previously reported; Instruction tuning increases novelty not just stylistically, but semantically.

**Weaknesses:**

Didn't compare results with baselines like InfiniGram

The choice of models is limited to one family and small sizes. It would be better to consider models from the OLMo family, as they are also fully open.

**Questions:**

Have you tried other embeddings and re-rankers?

Have you examined the effect of the in-context novelty itself on the novelty of the model output?

---

> ### Author Response · Authors · 2025-11-22
> **Author response**
>
> Thank you for your review. We address concerns and questions in the following:
>
> > **Didn't compare results with baselines like InfiniGram**
>
> While we agree that a quantitative comparison with n-gram-based baselines would be interesting, n-grams are unsuited for analyzing texts on the scale of hundreds of tokens, which is what inspired us to use semantic embeddings. This effect was shown by Rusty-DAWG (cf. Fig. 2/3 in their paper (Merill et al., 2024)): n-gram novelty approaches $1$ for as little as $n=23$ and completely converges to $1$ by $n=100$, as large n-gram overlaps are generally rare. We also personally verified this effect during our analysis and show a visual example in Fig. 1.
> Given these findings, both the human baseline and LLM curves for n-grams would have similarity values $\sim 0$ for all chunk sizes considered by us, and their quotient would carry too much noise to allow for a comparison with our approach, which is significantly more stable. It is therefore not reasonable to perform such an analysis.
>
> > **It would be better to consider models from the OLMo family, as they are also fully open.**
>
> We agree that cross-family validation is generally valuable, but would like to point out that the OLMo family is structurally similar to the 8 models that we already tested and its inclusion would be redundant for the purpose of our analysis:
> Both architectures are Llama-style architectures, utilize RMSNorm for normalization, SwiGLU for activation, and Rotary Positional Embeddings (RoPE) for context handling. Given this, OLMo does not offer a significant contrast to SmolLM and SmolLM2 beyond the architectural and training variance already covered in our study:
> - Data distribution shift: SmolLM is trained on less than 1T tokens on mainly synthetic data, while SmolLM2 is trained on 11T tokens with a focus on real data
> - Attention: Most models tested use standard multi-headed attention (MHA), while the SmolLM-360M models use grouped query attention (GQA)
> - Instruction tuning: The instruction-tuned versions of SmolLM2 were trained with DPO, while their counterparts in SmolLM were not
> - Context: SmolLM has a smaller context window and differently parameterized positional encodings than SmolLM2
>
> Based on the range of models we tested, we believe that adding OLMo to the analysis would increase computational cost significantly, while offering limited novel insights.
>
> > **Have you tried other embeddings and re-rankers?**
>
> We agree that investigating other embeddings and re-rankers would be useful. Unfortunately, performing ablations with various embedding models is infeasible at the scale of the pretraining corpus. However, we carefully picked GIST-large embeddings, which was among the top-10-performing embedding models for English text on the MTEB leaderboard at the time of the project, while having only 300M parameters and an embedding dimension of 1024. As such, it provides a strong trade-off between performance and feasibility. Moreover, we allow for slackness in the GIST ranking by saving the top-100 nearest neighbors of each query, and reranking with the significantly more powerful ColBERTv2 model, which is an established, strong model for re-ranking. This way, GIST only has to rank the true closest match within the top-100 for us to recover it later. Our ablation experiment in Appendix A substantiates the choice of $n=100$ for this step.
>
> > **Have you examined the effect of the in-context novelty itself on the novelty of the model output?**
>
> We are uncertain whether we understand the question. If we understand correctly, the question is whether the novelty score of the provided context in Fig. 3 correlates with the novelty score of the LLM continuation, if the LLM is prompted with that context.
>
> This effect is present and already observable in Fig. 3: The texts that are provided as context to the LLM are texts from the human baseline itself: For each document in our subset of Dolma, we provide the first 1000 tokens as context to the model, and then compare the model’s generation to the real continuation of that document. Therefore both the human baseline and the context come from adjacent parts of the same human-written document, i.e., from the same distribution. Fig. 3 illustrates that adding context from the human baseline draws the novelty values of the generations closer to the novelty values of the baseline.
>
> We hope that we understood the reviewer’s question correctly and are happy to answer any follow-up questions. Please let us know whether our understanding is correct.
>
> References:
>
> Merill et al. (2024): Evaluating n-Gram Novelty of Language Models Using RUSTY-DAWG.

---

### Official Review · Reviewer_AHZ9 · 2025-10-31

**Soundness:** 2
**Presentation:** 2
**Contribution:** 2
**Rating:** 4
**Confidence:** 2

**Summary:**

This paper introduces “un-attributability”: a new way of thinking about how novel model generations are in relation to the corresponding pre-training data. This approach involves indexing the entire corpus with GIST embeddings and for new model generations measuring semantic similarity (with some reranking). Any time a human reference ranks higher than retrieved chunks, that generation is deemed un-attributable. They run experiments on SmolLM and SmolLM2 with publicly available pre-training data and report findings on unprompted vs. prompted generations, fine-tuned models and for domain-specific tasks.

**Strengths:**

Thinking about what is not attributable seems interesting, though it might be hard to prove a negative.
Gist + faiss + colbert seems like a nice bit of infra to contribute to the community.

**Weaknesses:**

The discrepancy between the question: “which outputs cannot be attributed to any pretraining example” and then measuring semantic similarity to quantify this is confusing. What’s being measured here is less-constrained than vanilla token memorization but much weaker than TDA. Un-attributability seems like an over-claim here and since it is a key part of what is being proposed, it would be helpful to flesh this out.

**Questions:**

Could you clarify what the human references are and what kind of coverage they have in relation to the generations being evaluated for novelty?

---

> ### Author Response · Authors · 2025-11-22
> **Author response**
>
> We thank the reviewer for their questions and hope to clarify our reasoning in the following.
>
> > **The discrepancy between the question: “which outputs cannot be attributed to any pretraining example” and then measuring semantic similarity to quantify this is confusing. [...] Un-attributability seems like an over-claim here and since it is a key part of what is being proposed, it would be helpful to flesh this out.**
>
> We believe there are some parts of the paper that were unclear and leading to this confusion. We hope to clarify in the following, and have also updated the current revision to be more clear:
>
> As noted in our introduction, we follow the categorization in Chang et al. (2024) who distinguish between two branches of TDA: factual attribution and causal influence. When we speak of attributability or un-attributability in the remainder of the paper, we refer to the first notion, namely factual attribution, measured by a lexical or semantic overlap between generations and training data. This analysis can be done with BM25, n-grams, or in our case, embedding based similarity. The factual approach is more widely used today to analyze pretraining corpora (e.g., Liu et al. (2024), Liu et al. (2025)). Factual attribution is easier to scale to the full pretraining corpus and can provide valuable insights that do not rely on causality, such as questions about data contamination and memorization.
>
> As the reviewer pointed out, this is much weaker than causal influence, which establishes causal relationships between the training data and model outputs.
> We suggest that it is possible to re-introduce a stronger claim while sticking to factual attribution methods, by turning the question on its head: If a generated text is coherent and factually correct, but is not similar to any part of the training data, whether paraphrased or quoted, then we must conclude that the model inferred this information through logically combining facts seen during training. We call such a generation novel w.r.t. the training corpus, because the information doesn’t directly appear in the corpus, but needs to be inferred from the training data, e.g. through human reasoning or LLM training and inference. The presence of novel generations is direct evidence of model creativity, and generalization capabilities, so our goal is to quantify and study the novelty of model generations under varying conditions.
>
> We apologize for the confusion and remain available for any additional questions.
>
> > **Could you clarify what the human references are and what kind of coverage they have in relation to the generations being evaluated for novelty?**
>
> We choose a human baseline which is of the same length (via chunking) and the same domain as the LLM generation, and known to be novel w.r.t the corpus. As we know that the Dolma dataset is not part of SmolLM1 and SmolLM2, we use this dataset as the human references. Specifically:
> when we prompt an LLM to continue a Dolma document, we pick the real continuation of that document as a per-element baseline.
> for GSM8K and TruthfulQA, we pick the ground truth solution to that benchmark as a baseline, which is human-written
> for OpenRewriteEval, we pick randomly sampled texts from dolma as a baseline, because the ground truth solutions to this benchmark are themselves LLM-generated. The nature of the task (open-ended text generation and reformulation) is compatible with comparison to long-form human-written text
> Conceptually, as the human baseline is completely independent of the corpus, it is entirely non-attributable and its novelty does not change regardless of chunking. Hence, the scores that our retrieval pipeline attributes to the human baseline under varying conditions quantify unwanted LLM-independent effects, such as length bias, domain bias, and any potential natural-language-related trends.
> Therefore we report values relative to this baseline, isolating the LLM-specific novelty distribution and making the scores interpretable.
>
> We are happy to answer any follow-up questions.
>
> References:
>
> Chang et al. (2024): Scalable Influence and Fact Tracing for Large Language Model Pretraining.
>
> Liu et al. (2024): Infini-gram: Scaling Unbounded n-gram Language Models to a Trillion Tokens.
>
> Liu et al. (2025): OLMoTrace: Tracing Language Model Outputs Back to Trillions of Training Tokens.

---

### Official Review · Reviewer_tt4B · 2025-10-31

**Soundness:** 3
**Presentation:** 3
**Contribution:** 3
**Rating:** 6
**Confidence:** 4

**Summary:**

This paper studies language model generation un-attributability: which LM outputs are not similar to spans of text in the pretraining corpus? The paper proposes a two-stage embedding-based retrieval pipeline: first, candidate documents are retrieved via GIST-embedding cosine similarity, and then candidates are reranked using ColBERTv2. This second stage takes care to account for the chunk-based nature of the comparisons by considering multiple different chunk sizes. Experiments on SmolLM and SmolLM2 in open-domain settings show that generations continuing from a context are more novel than human baselines, among other findings. In domain-specific settings, the results show that smaller models yield  more novel generations than larger models, and novelty varies by task domain.

**Strengths:**

1. The paper operationalizes a new notion of un-attributability to study the novelty of LLM generations that goes beyond the surface form metrics used in prior work. It estimates novelty through a two-stage embedding retrieval pipeline to identify training documents that are similar to a generation.
2. The experiments are sound, and novelty values are contextualized against a human baseline that is known to be novel.
3. The experimental takeaways are interesting and help shed light on the relationship between training data and model generations, and I can imagine other approaches could build on the proposed notion of un-attributability.

**Weaknesses:**

1. More thorough evaluation of the retrieval pipeline would bolster the claims about novelty. For example, the claim "Novelty increases with sequence length in unprompted generation" (line 310 and Figure 3) could be true, but it could also be the case that retrieval gets worse with increasing sequence length. Can you provide evidence that the latter is not the case?

2. Novelty may not be enough on its own to understand language model outputs. Take the claim in line 378: "Smaller SmolLM2 models are more novel than larger ones." Novelty doesn't seem like it's useful on its own here---higher novelty could be explained by smaller models yielding disfluent text, which would be dissimilar from pretraining corpus text. Perhaps contextualizing novelty with model performance (or even qualitative examples of more novel and less novel outputs and their retrieved neighbors) would be useful.

3. The paper mentions that its results go beyond lexical methods for assessing generation novelty like [1] (lines 85-87). Providing concrete quantitative comparisons to findings from n-gram tools like Rusty-DAWG would strengthen the impact of this paper.

[1] Evaluating n-Gram Novelty of Language Models Using Rusty-DAWG, Merrill et al., EMNLP 2024.

**Questions:**

1. I really like the attention paid to chunking for stage 2 of the retrieval pipeline. Do you have any analysis for how much chunking boundaries can affect stage 1, embedding retrieval?

Small comments:
- There's a mismatch between human generation baseline values: it's defined as 1 in lines 242-243 but 0 in Figures 3 and 4.
- I suggest making the title more specific, perhaps even a small change like "Un-attributability: computing **generation** novelty from retrieval & semantic similarity"

---

> ### Author Response · Authors · 2025-11-22
> **Author response**
>
> We appreciate the encouraging review and thank the reviewer for uncovering the mistake in notation, which we fixed in the current version. We also agree with the title suggestion and adjusted it accordingly.
>
> In the following, we address the reviewer’s questions one by one:
>
> > **[...] it could also be the case that retrieval gets worse with increasing sequence length. Can you provide evidence that [this] is not the case?**
>
> This is in fact possible, and a major motivation for why we chose to report scores relative to a human baseline. If retrieval has any length bias, it would apply to both the baseline and the LLM-generated text. Therefore, whatever remains after taking the relative score, must reveal the actual novelty trend of the LLM.
>
> > **higher novelty could be explained by smaller models yielding disfluent text, which would be dissimilar from pretraining corpus text.**
>
> We understand your concern. It is true that our novelty measure cannot distinguish noisy or disfluent text from creatively written or generalized text. This is specifically why we performed our domain-specific analysis on generative benchmarks, as this allowed us to pick only the generations that constitute a correct solution to the given task. By constraining our analysis to correct samples, we mitigate the effect of disfluency on the reported novelty trends.
>
> > **Providing concrete quantitative comparisons to findings from n-gram tools like Rusty-DAWG would strengthen the impact of this paper**
>
> While we agree that a quantitative comparison would be interesting, n-grams are unsuited for analyzing texts on the scale of hundreds of tokens, which is what inspired us to use semantic embeddings. This effect was shown by Rusty-DAWG (cf. Fig. 2/3 in their paper (Merrill et al., 2024)): n-gram novelty approaches $1$ for as little as $n=23$ and completely converges to $1$ by $n=100$, as large n-gram overlaps are generally rare. We also personally verified this effect during our analysis and show a visual example in Fig. 1.
>
> Given these findings, both the human baseline and LLM curves for n-grams would have similarity values $\sim 0$ for all chunk sizes considered by us, and their quotient would carry too much noise to allow for a comparison with our approach, which is significantly more stable. We found it therefore not reasonable to perform such an analysis. We are happy to discuss any open concerns or questions.
>
>
> > **Do you have any analysis for how much chunking boundaries can affect stage 1, embedding retrieval?**
>
> We thank the reviewer for raising this point and ran additional experiments to analyze the effect of chunking on the embedding retrieval.
>
> Specifically, we sampled 10,000 documents of length 2,500-7,500 tokens from fineweb-edu and picked a sentence at the center of each document, of length 50-150 tokens. Using this setup, we compared various chunking borders, in which the target sentence would appear at different points within a chunk. Each time, we embed all chunks and the chosen sentence using GIST embeddings and compute cosine similarity between them, mimicking the procedure within our retrieval pipeline. For ranking to be successful, the chunk containing the sentence needs to have the highest cosine similarity with the embedding of the sentence, compared to other chunks of the text.
>
> We find that while the retrieval mechanism exhibits a bias towards early information appearance within a chunk, the ranking is at most affected by $1$. The median rank remains stable regardless of where the sentence appears, showing little effect of chunking. This is also true when chunking borders split the sentence, distributing it between two chunks. Crucially, we add an overlap of $50$ tokens to the chunking mechanism used in the final retrieval pipeline, and sample the top 100 ranks during the first retrieval stage, meaning that our final results are not affected by this finding.
>
> **We have added this analysis to Appendix B of our revision**. Currently, the preliminary results using 1,900 documents are shown, and will be replaced with the adjusted values for 10,000 documents within the next few days.
>
>
>
> References:
>
> Merill et al. (2024): Evaluating n-Gram Novelty of Language Models Using RUSTY-DAWG.

---

> ### Author Response · Authors · 2025-11-24
>
> We would like to inform the reviewer that we have uploaded a new revision of the paper, now containing the full analysis of chunk borders with 9518 documents tested in total. We are happy to answer any follow-up questions.

---

### Official Review · Reviewer_1DVE · 2025-11-01

**Soundness:** 1
**Presentation:** 2
**Contribution:** 2
**Rating:** 2
**Confidence:** 5

**Summary:**

The paper defines novelty as un-attributability: an LLM output is “novel” if the pretraining corpus contains no close semantic match under a two-stage retrieval pipeline—GIST+FAISS for top-n candidates, then ColBERTv2 for token-level re-ranking and scoring. Scores are length-normalized and calibrated against a human baseline (domain-matched text); the per-chunk similarity ratios are summarized by the median, and values < 1 are interpreted as “more novel than human baseline.”
The authors show experiments on SmolLM / SmolLM2 across open-ended and benchmarked settings (e.g., GSM8K, TruthfulQA, OpenRewriteEval). They report trends like domain-dependent novelty, smaller models sometimes appearing more “novel,” and instruction-tuned models showing lower similarity (hence higher “novelty”) than bases, all relative to the human baseline.

**Strengths:**

- Scalable technique: This recasts novelty as a retrieval problem that is feasible at pretraining scale; easy to implement and the pipeline and artifacts are open-sourced for reproducibility.


- Empirical results: domain effects and instruction-tuning effects on the similarity-to-corpus curves, with baselines are shown in terms of how finetuning affects novelty

**Weaknesses:**

- Semantic similarity is not causal attribution. While the authors acknowledge the method “is not a drop-in replacement” for causal attribution, it is quite contradictory to call it “un-attributable”. Could the authors clarify this ?

- The papers they cite in the paper such as [1] already show that TDA can be scaled to pretraining data, which already negates their fundamental premise of that TDA is not scalable


- The paper also assumes that failure to retrieve a neighbor is taken as novelty - this aspect is very much pipeline-dependent (embeddings, chunking, reranker). If the GIST embeddings aren’t accurate enough, the assumption that a chunk is novel might not hold.


- The paper also assumes that only “one” retrieved fact should be responsible for a generated text chunk. There are papers that challenge this assumption [2]. This makes the authors’ assumption weak, unless they rebut this strongly.


- No causal validation: There are no ablations (e.g., remove or paraphrase specific shards and re-measure) to connect the proxy to actual training-data influence, so findings remain correlational. The paper itself flags this limitation while also claiming this relates to unattributability


[1] Scalable Influence and Fact Tracing for Large Language Model Pretraining. Chang et al, 2024

[2] Simfluence: Modeling the Influence of Individual Training Examples by Simulating Training Runs. Guu et al, 2023

**Questions:**

- Baseline dependence - The “1.0” anchor moves with the chosen human baseline, dataset and chunk size; Can the authors provide a more detailed explanation how this works?

- What exactly is the motivation to choose the human baseline ? Why was it done this way?

---

> ### Author Response · Authors · 2025-11-22
> **Author response 1/2**
>
> Thank you for your thoughtful and thorough review. We hope to clarify concerns in the following.
>
> > **Semantic similarity is not causal attribution [...] it is quite contradictory to call it “un-attributable”. Could the authors clarify this?**
>
> Thank you for highlighting this. We follow the reviewer’s citation [1] who observe a “misalignment between factual *attribution* and causal *influence*.” (p.1) Hence when we speak of “un-attributability”, we refer to the notion of **factual attribution**, which inherently does not have a claim of causality. Yet, we argue that it is particularly interesting to study which LLM generations are *un-attributable* to the training corpus, as these indicate generation novelty. We realize that this distinction may not have been clear enough in the submission and emphasize it more strongly in the uploaded version.
>
> > **No causal validation: There are no ablations (e.g., remove or paraphrase specific shards and re-measure) to connect the proxy to actual training-data influence, so findings remain correlational.**
>
> We apologize for the unclarity and resulting confusion between our use of the “attribution” terminology. As explained in the answer to the previous point, we use the term *attribution* in the sense of factual attribution, and do not claim the discovery of causal influence. The focus of our study is to discover and analyze when LLM generations are novel compared to the training corpus, for which we leverage attribution.
>
> > **The papers they cite in the paper such as [1] already show that TDA can be scaled to pretraining data, which already negates their fundamental premise of that TDA is not scalable.**
>
> We believe there may be a miscommunication, and would like to clarify: Our work does not assume that causal TDA is in general not scalable, as [1] also demonstrates. Rather we emphasize that, compared to factual attribution approaches, causal attribution is harder to scale as it is computationally more expensive. At the same time, we would like to highlight that **the question of un-attributability does not require causal influence**: We can make use of factual attribution pipelines to study generation novelty.
>
> > **The paper also assumes that failure to retrieve a neighbor is taken as novelty - this aspect is very much pipeline-dependent (embeddings, chunking, reranker). If the GIST embeddings aren’t accurate enough, the assumption that a chunk is novel might not hold.**
>
> We agree, and this is indeed the major limitation of our paper. Unfortunately, performing ablations with various embedding models is infeasible at the scale of the pretraining corpus. However, we carefully picked GIST-large embeddings, which was among the top-10-performing embedding models for English text on the MTEB leaderboard at the time of the study, while having only 300M parameters and an embedding dimension of 1024. As such, it provides a strong trade-off between performance and feasibility. Moreover, we allow for slackness in the GIST ranking by saving the top-100 nearest neighbors of each query, and reranking with the significantly more powerful ColBERTv2 model. This way, GIST only has to rank the true closest match within the top-100 for us to recover it later. Our ablation experiment in Appendix A substantiates the choice of $n=100$ for this step.
>
> > **The paper also assumes that only “one” retrieved fact should be responsible for a generated text chunk [...]**
>
> We believe there may be a misunderstanding of our reasoning behind un-attributability and would like to clarify:
> As reference [2] shows, the above assumption is challenged in the current literature. This is why **we do not make this assumption in our paper**. We instead consider the opposite notion: if the corpus does not contain a generated fact *at all*, then it has to have been inferred by the LLM during the training process in some way, showing that the model generalized beyond the training data.
> While it is true that causal influence can be computed for all generations, we specifically call factually non-attributable generations novel because they require the model to generalize beyond the training data to create these new ones. In contrast, non-novel generations, as we define them, are mere reproductions of the training data.
>
> To further clarify, we provide an example:
> Let the corpus contain the two statements “plants perform photosynthesis” and “trees are plants”, then we define the generation “trees perform photosynthesis” as novel under our framework. While causal attribution can track the generation back to the corpus, the generated fact itself was not part of the training data. Therefore, compared to the training data, it is a new piece of information that could only have been produced through understanding of the underlying data.
>
> We hope that this brings some clarity into what we mean by novelty and un-attributability. We are happy to discuss any follow-up questions.

---

> ### Author Response · Authors · 2025-11-22
> **Author response 2/2**
>
> > **Baseline dependence - The “1.0” anchor moves with the chosen human baseline, dataset and chunk size; Can the authors provide a more detailed explanation how this works? What exactly is the motivation to choose the human baseline ? Why was it done this way?**
>
> We thank the reviewer for raising this confusion and hope to clarify: We compare against a human baseline in generation novelty to enable us to interpret our results.
>
> When using raw similarity scores, two problems arise: (1) As neither GIST nor ColBERTv2 are interpretable models, their semantic similarity scores are non-interpretable by design. (2) Moreover, similarity scores can have trends arising from the nature of language itself: Merill et al. (2024) show that the frequency of n-grams within model generations that are also present in the corpus forms a sigmoid-like curve and converges to $1$ at $n \sim 100$. It is reasonable to assume that our scores, too, have a natural trend, and they additionally inherit domain or length biases from the models that we use.
>
> For example:
> If raw similarity scores drop with larger chunk sizes, then in absence of a baseline it would be impossible to know whether this effect is due to the LLM, due to a length bias in the retrieval pipeline, or due to the structure of natural language. We are however interested in isolating the contribution of the LLM. We are also interested in making the drop of similarity scores interpretable: what would it mean to go from $0.7$ to $0.6$?
>
> We choose a human baseline which is of the same length (via chunking) and the same domain as the LLM generation, and known to be novel w.r.t the corpus. For instance, when we prompt an LLM to continue a Dolma document, we pick the real continuation of that document as a per-element baseline.
> Conceptually speaking, **as the human baseline is completely independent of the corpus, it is entirely non-attributable**: Its novelty doesn’t change regardless of chunking. Hence, the scores that our retrieval pipeline attributes to the human baseline under varying conditions precisely quantify the aforementioned unwanted effects.
> Reporting values relative to this baseline now solves both problems at once: it (1) isolates the LLM-specific novelty trend, and (2) it makes the scores interpretable: a drop from $0.7$ to $0.6$ now means that the first generation is 30% more novel than comparable human text, while the second generation is 40% more novel.
>
> We hope that this response answered the reviewer’s questions and are happy to answer any follow-up questions.
>
> References:
> Merill et al. (2024): Evaluating n-Gram Novelty of Language Models Using RUSTY-DAWG.

---

> ### Comment · Reviewer_1DVE · 2025-11-25
>
> - Thanks for the clarification on attributability. I think the assertions in the paper are more clear now.
>
> - GIST embeddings: Thank you for clarifying this. But, my fundamental issue still holds. I personally think this paper is better framed as a study on GIST embeddings. It studies the question of "how effective is GIST embeddings for factual attribution" and "novelty as a function computed via GIST embeddings"
>
> - multiple facts: I think there was a misunderstanding here. Assume this setup: Lets says a test fact is "A rel C" can be inferred from a multihop statements "A rel_1 B". "B rel_2 D" and "D rel_3 C". And its natural to expect an embedding retrieval to bring facts with "A" and "C" indepedently and rank them independent of whether "B" and "D" were present. This is the assumption I was challenging. That remains unaddressed
>
> - (rest of my questions are clarified)
>
> I still believe this paper would benefit from a round of revision (I still like this paper and want to see it appear to the right audience). Rather than the stated claims, I view this paper is a meaningful study on GIST+ColBERT, but the broader claims and hypotheses this paper makes is where the primary weakness is. I maintain my score for now.

---

> ### Author Response · Authors · 2025-11-28
>
> - Regarding GIST embeddings: Thank you for the feedback. We understand your perspective. We believe that the discussion has overly emphasized the specific choice of embedding model, whereas our primary contribution is the proposal to study novelty as *unattributability* with embeddings. In retrospect, we placed too much weight on the embedding model choice in the paper, although it is ultimately a secondary design decision, and we apologize for the confusion this caused.
> Our main goal is to analyze which factors (like model size, chunk size, or instruction tuning) affect the novelty of LLM generations relative to their training corpus, evaluated over a query set of multiple generations. For this analysis, the particular embedding model is largely irrelevant as long as it performs reasonably well: all novelty comparisons are made between texts from the same domain and with comparable length. Even if an embedding model retrieved only ~80–90% of queries correctly, a consistent gap in novelty scores between two LLMs would still meaningfully support our conclusions.
> For practical reasons, and to reduce measurement noise, we selected GIST because it was among the top-performing embedding models available at the time. However, our methodology and core claims do not depend on this specific choice.
>
> - Regarding multiple facts: Thank you for clarifying. As far as we understand, we are indeed assuming that a generation is attributed to singular, independent, semantically similar documents in the corpus, irrespective of their position within potential multihop statements. Please let us know whether our understanding is correct. In the example of “A rel C,” semantic attribution would typically include facts related to both A and C, with the closest matches being other instances of the explicit relation “A rel C.” and miss facts on "B rel D". However we would like to emphasize that our focus is not on what is attributable, but rather on what is not. Our goal is to determine whether the model is simply reproducing its training set or inferring new knowledge from it.
> The multihop statement you describe requires integrating information across documents and exercising generalization abilities. These are precisely the kinds of cases we classify as novel, and whose occurrence we analyze, because they provide evidence of deeper understanding and logical reasoning over the data. The assumption about attributability is specifically what is necessary to detect such cases.

---

### Author Response · Authors · 2025-11-22
**General Author Response**

We would like to sincerely thank the reviewers for their thoughtful and constructive feedback. The reviews are very helpful in significantly improving the clarity and quality of the paper.

We have already incorporated much of the suggestions and updated the paper to clarify the open questions and upload the new revision with this response. The updates are **highlighted in blue** for ease of review.

Reviewers requested additional analysis (effect of chunking on stage 1 retrieval), which we agree with and have added to the paper (Appendix B). The experiments show that while different chunking strategies can lead to different rankings in the first stage of the pipeline, these ranking differences are minimal (i.e., at most switching among the first ranks). Moreover, the differences **do not affect the pipeline** as we retrieve the top 100 ranks and rerank them in stage 2.

---

> ### Author Response · Authors · 2025-11-24
>
> We have updated the paper once more, to include the final results of the additional analysis in Appendix B, which now includes 9518 tested documents.

---

### Note · Authors · 2026-01-04

I have read and agree with the venue's withdrawal policy on behalf of myself and my co-authors.